# Catalytic and Stoichiometric *Baeyer–Villiger* Oxidation Mediated by Nonheme Peroxo-Diiron(III), Acylperoxo, and Iodosylbenzene Iron(III) Intermediates

**DOI:** 10.3390/molecules27092814

**Published:** 2022-04-28

**Authors:** Dóra Lakk-Bogáth, Miklós István Szávuly, Patrik Török, József Kaizer

**Affiliations:** Research Group of Bioorganic and Biocoordination Chemistry, Faculty of Engineering, Center for Natural Sciences, University of Pannonia, H-8201 Veszprém, Hungary; lakkd@almos.uni-pannon.hu (D.L.-B.); miklos_szavuly@yahoo.com (M.I.S.); patriktrk6@gmail.com (P.T.)

**Keywords:** *Baeyer–Villiger* oxidation, peroxo-diiron(III), oxoiron(IV), catalysis, kinetic studies

## Abstract

In this paper we describe a detailed mechanistic studies on the [Fe^II^(PBO)_2_(CF_3_SO_3_)_2_] (**1**), [Fe^II^(PBT)_2_(CF_3_SO_3_)_2_] (**2**), and [Fe^II^(PBI)_3_](CF_3_SO_3_)_2_ (**3**)-catalyzed (PBO = 2-(2′-pyridyl)benzoxazole, PBT = 2-(2′-pyridyl)benzthiazole, PBI = 2-(2′-pyridyl)benzimidazole) *Baeyer–Villiger* oxidation of cycloketones by dioxygen with cooxidation of aldehydes and peroxycarboxylic acids, including the kinetics on the reactivity of (μ-1,2-peroxo)diiron(III), acylperoxo- and iodosylbenzene-iron(III) species as key intermediates.

## 1. Introduction

*Baeyer–Villiger* oxidation has become one of the most important reactions in organic chemistry with a large range of possible applications because the forming lactones or esters are important industrial intermediates in the synthesis of various pharmaceuticals, monomers for polymerization, pheromones, and herbicides for agrochemistry. These reactions have been commonly carried out by the use of expensive, shock-sensitive, and potentially explosive peroxycarboxylic acids as oxidants [1,2,3,4]. Peracids (iminoperacids) can be efficiently generated in situ from nitriles (solvent) and H_2_O_2_ in the presence of solid bases or from aldehydes and dioxygen in the presence of metal compounds [5,6,7,8,9,10,11,12,13,14,15,16,17]. This way, the major disadvantages, namely handling large amounts of peracid, and the noncatalytic use of acid, can be avoided. A combination of molecular oxygen and aldehydes under homogeneous and heterogeneous catalysis has also been extensively studied. *Baeyer–Villiger* oxidation of cyclic ketones to lactones catalyzed by silica-supported nickel complex, iron(III)-containing mesoporous silica, and Mg-Al hydrotalcites has been reported [18,19,20,21,22,23,24]. In the homogeneous phase, efficient selective oxidation of cyclohexanones to lactones by molecular oxygen with benzaldehyde as an oxygen acceptor, in the presence of Fe(TPP)Cl, Ru(TPP)Cl, Co(TPP)Cl and Mn(TPP)Cl (TPP—meso-tetraphenylporphyrin) complexes, has been reported [25,26]. In the benzaldehyde-mediated aerobic *Baeyer–Villiger* oxidation of the cyclohexanone system, high-valent iron porphyrin was the oxidative species to produce 3-caprolactone [26]. Type III *Baeyer–Villiger* monooxygenases (BVMOs [27,28,29,30,31,32,33]) are specific cytochrome P450s, which are involved in the synthesis of brassinosteroids—steroidal hormones essential for the growth and development of plants [27]. Iron complexes of *meso*-tetraphenylporphyrin, Fe^III^(TPP)Cl [25,26], and *N*,*N*-bis(2-pyridylmethyl)-*N*-bis(2-pyridyl)methyl-amine [Fe^II^(N4Py)(CH_3_CN)]^2+^, proved to be efficient catalysts for the aerobic oxidation of cyclohexanone in the presence of various aldehydes as sacrificial reductants, wherein, contrary to the heme-containing monooxygenases, a high-valent iron porphyrin, [Fe^V^(TPP)(O)]Cl and [Fe^IV^(N4Py)(O)]^2+^ were proposed as key intermediates in the rate-determining oxygen atom transfer step to generate the *ε*-caprolactone [34]. Oxidation of four substituted cyclohexanone derivatives by oxoiron(IV) complex, [Fe^IV^(asN4Py)(O)]^2+^ with chiral pentadentate ligand, *N*,*N*-bis(2-pyridylmethyl)-1,2-di(2-pyridyl)ethylamine, attained moderate enantioselectivities up to 45% enantiomeric excess (*ee*) [35]. In the literature, only a few examples can be found where the peroxo-diiron(III) species is capable of direct nucleophilic reactions, such as oxidative deformylation of aldehydes and *Baeyer–Villiger* oxidation of cyclohexanones [36,37,38,39]. The coordination chemistry of nitrogen-rich nonsymmetric bidentate ligands has received much attention. We have demonstrated that by tuning the ligands σ-donor/π-acceptor strength, the reactivity and the formation rate of intermediate (μ-1,2-peroxo)diiron(III) complexes in the reaction of their iron(II) precursor complexes with H_2_O_2_ can be influenced [38]. Here we report the formation of (μ-1,2-peroxo)diiron(III), Fe^III^(mCPBA), and Fe^III^(OIPh) complexes by the use of various oxidants such as H_2_O_2_, *m*-chloroperoxybenzoic acid (mCPBA), iodosylbenzene (PhIO), and benzaldehyde with O_2_. and their nucleophilic reactivity in stoichiometric and catalytic *Baeyer–Villiger* oxidation reactions (Figure 1).

## 2. Results and Discussion

We have previously reported the synthesis and structure of [Fe^II^(PBO)_2_(CF_3_SO_3_)_2_] (**1**), [Fe^II^(PBT)_2_(CF_3_SO_3_)_2_] (**2**), and [Fe^II^(PBI)_3_](CF_3_SO_3_)_2_ (**3**) (PBO = 2-(2′-pyridyl)benzoxazole, PBT = 2-(2′-pyridyl)benzthiazole, PBI = 2-(2′-pyridyl)benzimidazole) complexes, and spectroscopic characterization of their transient green species with a Fe^III^(μ-1,2-O_2_)Fe^III^ core (λ_max_ = 685–720 nm, and ε ~1400) as a result of the reaction of **1**–**3** with H_2_O_2_ [40]. Based on detailed kinetic and computational studies, we have found direct evidence for the formation of low-spin oxoiron(IV) species in a pre-equilibrium process during the oxidation of phenols as ribonucleotide reductase (RNR-R2) models [41]. We have also published the iron(II)-catalyzed *Baeyer–Villiger* oxidation of cyclohexanone, where similarly to the previously published Fe^III^(TPP)Cl [25,26] system, oxoiron(IV) intermediate was suggested as a reactive intermediate during the oxygen transfer step. Since the two intermediates above have different characteristics (electrophilic versus nucleophilic for Fe^IV^O and Fe^III^(μ-1,2-O_2_)Fe^III^, respectively), the question arises as to which form can be used to interpret the mechanism in the case of our selected complexes (**1**–**3**). Since the mechanism of the *Baeyer–Villiger* reaction can be interpreted essentially through a nucleophilic addition (A_N_) step, the peroxo-diiron intermediate may be a suitable candidate. Our primary goal is to elucidate the role of the two possible intermediates in the catalytic and stoichiometric oxidation reaction of cycloketones.

### 2.1. Catalytic Tests for the Iron(II)-Catalyzed Baeyer–Villiger Oxidation of Cycloketones

As a first step, the catalytic activity of complexes **1**–**3** was investigated using the conditions described in the literature for the Fe^III^(TPP)Cl-containing catalytic system [25,26]. Reactions were carried out in toluene at 60 °C under air, where catalyst, substrate, and aldehyde were in a molar ratio of 1:1000:15,000, respectively (Table 1 and Figure 1). In this system, peracids can be efficiently generated in situ from aldehydes and dioxygen in the presence of metal compounds, which act as the active oxygen species in the B–V reaction. The consumption of the cyclohexanone and the formation of the ε-caprolactone were monitored by GC and GC-MS. There was, remarkably, a difference in efficiency between the three kinds of catalysts (**1**–**3**) compared with the previously investigated Fe^III^(TPP)Cl complex. Figure 1 shows the profiles of cyclohexanone aerobic oxidation catalyzed by PBO, PBT, and PBI complexes with benzaldehyde as coreductant, compared with the previously reported metalloporphyrin-catalyzed B–V oxidation system. The conversion of cyclohexanone in all cases increased rapidly within the first 1 h period (Figure 1a), and the conversion reached 48%, 72.8%, and 85% after 5 h for **3**, **2**, and **1**, respectively. The relative reactivity of catalysts is in the following order **1** > **2** > **3** (Figure 1a). The obtained reactivity order can be explained by the different structures of the complexes and the effect of the ligand framework. Much higher reactivity was observed for the coordinative unsaturated bisz Fe^II^(PBO)_2_ (**1**) and Fe^II^(PBT)_2_ (**2**) complexes. Furthermore, the electrochemical properties of the complexes show significant differences, which may also explain the different reactivity. Complex **3** exhibit a quasi-reversible redox couple at 0.90 V vs. Ag/AgCl (*E*_pa_(Fe^III/II^) = +0.94 V and *E*_pc_(Fe^III/II^) = +0.85 V). The irreversible reductions at potentials more negative than −1.0 V are assigned to ligand-centred one-electron reductions. The Fe^III^/Fe^II^ redox couples of **1** (*E*_pa_(Fe^III/II^) = +1.55 V and *E*_pc_(Fe^III/II^) = +0.42 V) and **2** (*E*_pa_(Fe^III/II^) = +1.44 V and *E*_pc_(Fe^III/II^) = +0.32 V) are both irreversible and are considerably higher potentials than for **3**, consistent with the electron-withdrawing nature of O and S compared with NH [40].

It can be seen that the conversions increases with time in all cases (Table 1). However, catalyst efficiency (turnover frequency (TOF) = the number of turnovers/h) decreased with time, indicating that with longer reaction times, catalytic efficiency could be lost.

Since the solvent can play a role in the stabilization of polar intermediates during the reaction pathway, in this sense, acetonitrile with a higher polarity was chosen as the solvent. Based on our previous experience, the most common solvent for the preparation of oxoiron(IV) and peroxo-diiron(II) intermediates is acetonitrile. Table 2 and Figure 2 present the preliminary kinetic results of the cyclohexanone oxidation catalyzed by **2** with benzaldehyde under oxygen, including the values of conversions and the number of turnovers. The conversion of cyclohexanone is 28% without catalyst but occurs with much higher yields in the presence of **2** compared with the classical *Baeyer–Villiger* reaction. The effect of the complex **2** concentration was investigated under fixed conditions (Table 2, entries 1–5, and Figure 2a) at 60 °C. High selectivity was achieved in all runs, and maximum conversions were obtained in the range of 0.01–0.10 × 10^−5^ M complex (**2**) concentration. It means that complex **2** proved to be an efficient catalyst for cyclohexanone oxidation. Further increase in the concentration of the complex results in a decrease in conversion, which can be explained by the oxidation of benzaldehyde as a competing substrate and the formation of a catalytically inactive μ-oxo-diiron(III) complex. However, in the O_2_/aldehyde oxidation system, large amounts of aldehydes were required as sacrificing agents for the oxidation of cyclohexanone to obtain high conversion values (Table 2, entries 6–10, Figure 2b).

Competitive reactions were also performed with parasubstituted benzaldehyde derivatives in order to evaluate the influence of electronic factors on the metal-free and metal-based reactions (Appendix A). Relative reactivities have shown linear correlations with Hammett’s σ constants. The negative reaction constants ρ were negative (ρ = −0.46 for **2** and −0.68 for BA/O_2_), suggesting that the rate-determining steps are nucleophilic in both cases. 

The scope of substrates for the *Baeyer–Villiger* oxidation catalyzed by the [Fe(PBT)(OTf)_2_] (**2**) was examined, and the typical results are shown in Table 3. In general, the more electron-rich (most-substituted) alkyl group migrates in preference but based on the calculated TOF values (~34), no significant effect has been observed for the alkyl substitution except for 4^t^Bu-cyclohexanone and 3-Me-cyclohexanone, probably because of a solubility problem and more sensitive steric 1,3-interactions, respectively (Figure 3, Table 3, entries 3 and 4). The same trend has been observed for the metal-free system indicating a similar mechanism.

Since the proposed oxidant is the peroxybenzoic acid (PBA) in the BA/O_2_ system studied above, we have also investigated the *Baeyer–Villiger* oxidation of cyclohexanone by the use of mCPBA as the oxidant. Figure 4 and Table 4 show the catalytic activity of the three catalysts (**1**–**3**). Among the catalysts tested, similarly to the BA/O_2_ systems, **2** and **3** showed the highest activity with ~70% conversion and ~100 turnover per hour.

The conversion of the cyclohexanone into ε-caprolactone can be significantly increased by increasing the amount of catalyst, and the highest conversion value (67%) was observed in a molar ratio of 1 (**2**):1000 (Substrate):15,000 (mCPBA) (Figure 5a and Table 5, entries 1–5). A similar ratio (1:1000:15,000) and conversion value (78%) were observed when the effect of oxidant was investigated (Figure 5b and Table 5, entries 6–9).

### 2.2. Stoichiometric Peroxo-Diiron(III)-Mediated Baeyer–Villiger Oxidation of Cycloketones

To get more insight into the mechanism of the catalytic reactions above, it was important to study the formation of possible intermediates by the use of various oxidants and investigate their stoichiometric oxidation with cycloketones. We have found earlier that the addition of H_2_O_2_ to acetonitrile solutions of the [Fe^II^(PBI)_3_](CF_3_SO_3_)_2_ (**3**) results in the rapid colour from red to green (λ_max_ = 720 nm, ε = 1360 M^−1^ cm^−1^), which can be ascribed to the charge transfer between Fe(III) and the O_2_^2−^ ligand [40,41]. Complex (**3**) can also be easily oxidized with mCPBA (Figure 6a), PhIO (Figure 7a), and BA under air (Figure 8a), resulting in a characteristic shift of the NIR absorption band in λ_max_ to 760 nm (ε= 1400 M^−1^ cm^−1^). The half-lives (*t*_1/2_’s) for complex **3^PhIO^** is 7200 s at 15 °C. Based on the UV–Vis spectra, intermediates **3^PhIO^**, **3^mCPBA^,** and **3^BA^** show a high degree of similarity to species **3^H_2_O_2_^**. These results may suggest the formation of metastable peroxodiiron(III) species in all cases.

A solution of **3** in MeCN was titrated with *m*CPBA dissolved in MeCN. Aliquots of *m*CPBA were added to the solution of **3**, and the UV–Vis spectral changes were recorded after each addition (Figure 6b). Correction for dilution was applied. Spectral changes at 760 nm were plotted against the added *m*CPBA. Almost the same species could be observed with complexes **1** and **2** but in much lower yields (~10% based on **3^PhIO^** at 5 °C), which can be explained by the much lower thermal stability of the forming intermediates caused by the two available coordination sites in the precursor bis-complexes. 

Similarly, a MeCN solution of **3** was subjected to titration with PhIO dissolved in CH_2_Cl_2_ (Figure 7b). Complex **3** reacts rapidly with *m*CPBA or PhIO at room temperature to afford **3^mCPBA,^** which is almost identical to **3^PhIO^**, as confirmed by UV–Vis. Consistent with the **3^mCPBA^**, titration of **3** with *m*-CPBA, monitored by UV–Vis spectroscopy, requires one equivalent of mCPBA. Similar changes are observed during the titration of **3** with PhIO. These results suggest a high similarity between **3^mCPBA^** and **3^PhIO^** species. 

We have previously reported that the rR spectroscopy at λ_exc_ 785 nm shows enhancement of bands at 876 and 463 cm^−1^ that are typical of a Fe(III)-O-O-Fe(III) core upon addition of H_2_O_2_ to the solution of **3 [40]**. The correspondence of the observed and calculated shifts in the bands at 876 cm^−1^ (to 826) and 463 cm^−1^ (to 445 cm^−1^) where H_2_^18^O_2_ was employed supported the assignment of bands as the O-O and Fe-O stretching modes, respectively. Despite the similarity of the UV–Vis spectra, the formation of peroxo-diiron species can be ruled out in the case of **3^mCPBA^** and **3^PhIO^** intermediates, based on their rRaman spectra (Appendix A). In the case of PhIO, the bands at 462 and 900 cm^−1^, which seem promising, are unfortunately not ^18^O sensitive, and the same species is formed regardless of oxidation. They are probably derived from PhI. Contrary to the EPR spectrum of **3^H_2_O_2_^**, which shows only trace levels of mononuclear high and low-spin iron complexes, the EPR signals of **3^PhIO^** and **3^mCPBA^** (g = ~2.29 and ~1.87) can be assigned to the S = ½ low-spin monomeric iron(III) species (Fe^III^(mCPBA) and Fe^III^(OIPh)) (Appendix A). Assignment of the EPR features of **3^mCPBA^** to a low spin 3-chloroperoxybenzoatoiron(III) complex is consistent with similar EPR features observed for other S = ½ low-spin acylperoxoiron(III) complexes [42].

Similar changes are observed during the titration of **3** with PhIO. These results suggest the formation of metastable peroxodiiron(III) species in both cases.

Figure 8a shows the formation of the in situ formed **3^PBA^** species in the reaction of **3** with an excess of benzaldehyde under air at 5 °C in CH_3_CN. The same species can be observed by the use of parasubstituted benzaldehydes under identical conditions. However, the resulting species decomposes rapidly, which can be explained by its reaction with excess benzaldehyde. The second-order rate constant in the oxidation of benzaldehyde with **3^PBA^** is 1.4 M^−1^s^−1^ at 298 K, which is twice less than that observed for complexes **3^H_2_O_2_^** (2.39 M^−1^s^−1^ at 288 K). The small difference in reaction rates may be explained by the different nature of the intermediates (**3^H_2_O_2_^** and **3^PBA^**) formed.

The Hammett plot analysis shows that the rate constant for the oxidation of benzaldehyde by the in situ-forming **3^PBA^** is sensitive to changes in the electronic properties of the benzaldehyde, with a ρ value of +0.43, suggesting a nucleophilic attack of the proposed peroxide on the aldehyde C-atom in the rate-determining step (8b). This result is consistent with that obtained for the two catalysed oxidation of cyclohexanone using parasubstituted benzaldehydes (Appendix A). Similar values were obtained for (μ-1,2-peroxo)diiron(III) complexes with Me-PBI (+0.67), and (μ-oxo)(μ-1,2-peroxo)diiron(III) complex with indH (+0.48) ligands [36,37,38,39].

To get direct evidence for the involvement of a **3^PhIO^** species in the *Baeyer–Villiger* oxidation, the reaction of **3^PhIO^** with various cycloketone derivatives was investigated. The **3^PhIO^** complex was generated by the reaction of **3** with PhIO, and the rate of the decay of the absorption band at 760 nm with cyclohexanones was measured as a function of the concentration of added cyclohexanone derivatives (Figure 9a). It was found that the **3^PhIO^** species is able to oxidize the cyclohexanone derivatives to the corresponding ε-caprolactones. The relative reactivity of substrates is in the following order: 4^t^Bu-cyclohexanone > cyclohexanone > 2Me-cyclohexanone > 3Me-cyclohexanone > 4Me-cyclohexanone (Figure 9a and Table 6). The oxidation of other cyclic ketones, such as cyclopentanone and cyclobutanone, was also examined (Figure 9b and Table 6). Their relative reactivity shows the following order: cyclohexanone > cyclopentanone > cyclobutanone, and correlates very well with their endocyclic bond angles (Figure 10). Since no reaction has been observed for benzophenone, this indicates clearly that the conjugation of the carbonyl group decreases the reactivity of the ketone.

The traces could be fitted with a first-order kinetic law, with respect to **3^PhIO^**, and the calculated *k*_ox_ values (-d [**3^PhIO^**]/d*t* = *k*_obs_[**3^PhIO^**] = (*k*_0_ + *k*_ox_[S])[**3^PhIO^**]), for different concentrations of the appropriate substrate are reported in Table 7. Kinetic experiments revealed first-order dependence on both the substrate (cyclohexanone) (Figure 11) and the **3^PhIO^** concentration (Figure 12a) with *k*_2_ = 7.17 × 10^−2^ M^−1^ s^−1^, Δ*H^≠^* = 23 ± 3 kJ mol^−1^, and Δ*S^≠^* = −185 ± 10 J mol^−1^ K^−^ at 15 °C (Figure 12b). This value is six times smaller than that measured for **3^H_2_O_2_^** (0.4 M^−1^ s^−1^) under the same conditions, which may also indicate a different structure of the two oxidants.

The low activation enthalpies and the large negative activation entropies are typical of associative processes. Almost the same values were observed for the (μ-oxo)(μ-1,2-peroxo)diiron(III)-mediated *Baeyer–Villiger* reaction (Δ*H^≠^* = 22 ± 1 kJ mol^−1^ and Δ*S^≠^* = −170 ± 10 J mol^−1^ K) [39].

In view of the kinetic and spectroscopic results obtained for the co-oxidants used, three different kinds of reaction mechanisms can be proposed (Figure 2). Based on our previously published results with benzaldehydes [36,39], the *Baeyer–Villiger* reaction is likely to occur through the μ-1,2-peroxo-diiron(III) intermediate by the use of H_2_O_2_ as cooxidant (Figure 2A). Since the formation of the peroxo species is fast, the rate-determining step is its reaction with the appropriate carbonyl compounds in a nucleophilic addition reaction (A_N_). Similar mechanisms can be proposed for **3^PhIO^**, **3^mCPBA^**, and **3^BA/O_2_^** containing systems, where, based on EPR and rRaman measurements, nucleophilic Fe^III^(OIPh) and Fe^III^(OO(O)CPh) adducts can be deduced as key oxidants (Figure 2B). It should be noted, however, that based on our previous and current results, we have found no evidence for the formation of characteristic oxoiron(IV) species [34] and their possible role in the oxidation of cyclohexanone via electrophilic OAT mechanism (Figure 2C).

## 3. Experimental Section

### 3.1. Materials and Methods

All syntheses were performed under an argon atmosphere unless stated otherwise. Solvents used for the synthesis and reactions were purified by standard methods and stored under argon. The starting materials for the ligand are commercially available, and they were purchased from Sigma-Aldrich (Budapest, Hungary). The ligands 2-(2′-pyridyl)benzimidazole (PBI), 2-(2′-pyridyl)benzthiazole (PBT), 2-(2′-pyridyl)benzoxazole) (PBO), and their complexes [Fe^II^(PBO)_2_(CF_3_SO_3_)_2_] (**1**), [Fe^II^(PBT)_2_(CF_3_SO_3_)_2_] (**2**), and [Fe^II^(PBI)_3_](CF_3_SO_3_)_2_ (**3**), were prepared as previously described [40]. Microanalyses were conducted by the Microanalytical Service of the University of Pannonia. The UV-Visible spectra were recorded on an Agilent 8453 diode-array spectrophotometer using quartz cells. GC analyses were performed on an Agilent 7820A (Budapest, Hungary) gas chromatograph equipped with a flame ionization detector and a 30 m HP-5 column. GC-MS analyses were carried out on Shimadzu QP2010SE (Budapest, Hungary) equipped with a secondary electron multiplier detector with conversion dynode and a 30 m HP-5MS column. Raman and EPR spectra were recorded at λ_exc_ 785 nm using a Perkin Elmer Raman Station at room temperature and Bruker ECS106 spectrometer in liquid nitrogen (77 K), respectively.

### 3.2. Catalytic Oxidations and Determination of Products

All reactions were carried out in a 20 mL Schlenk tube equipped with a condenser. Cyclohexanone (1.00 × 10^−2^ M), complex (1.00 × 10^−5^ M), acetonitrile (5 mL), and the initiator benzaldehyde derivatives or *m*CPBA (1.50 × 10^−1^ M) were added, and then the mixture was stirred at 60°C under an oxygen atmosphere 5–15 h. Unfortunately, the use of PhIO under catalytic conditions was not technically feasible because of solubility problems. The products were identified by GC (Agilent 7820A) and GC-MS (Shimadzu QP2010SE), and yields and conversions were calculated based on the amount of cyclohexanone consumed and products formed in the reactions using bromobenzene as an internal standard. High selectivity was achieved in all runs, and the calculated yields and conversions were almost identical (<5%).

### 3.3. Stoichiometric Oxidations

Complex **3** (0.5–2.0 × 10^−3^ M) was dissolved in acetonitrile (1.5 mL), then 4 equivalents of mCPBA or PhIO (or BA under air) were added to the solution. Cyclohexanone (0.1–0.35 M) was added to the solution, and the reaction was monitored with UV–Vis spectrophotometer (Agilent 8453, Budapest, Hungary) at 760 nm (ε = 1360 M^−1^ cm^−1^). The *Baeyer–Villiger* products (lactones) were identified by GC (Agilent 7820A) and GC-MS (Shimadzu QP2010SE).

## 4. Conclusions

In conclusion, we previously found that N4Py-based iron(II) complexes are capable of carrying out *Baeyer–Villiger* oxidation of cycloketones via the formation of oxoiron(IV) intermediate [34]. As a continuity of this study, efforts have been made to enhance the catalytic activity by the use of *n*–heterocyclic ligands and investigate the effect of the ligand framework. Comparing the reactions of [Fe^II^(PBO)_2_(CF_3_SO_3_)_2_] (**1**), [Fe^II^(PBT)_2_(CF_3_SO_3_)_2_] (**2**), and [Fe^II^(PBI)_3_](CF_3_SO_3_)_2_ (**3**) towards cyclohexanone under the same conditions, the relative reactivity is in the order of **1** > **2** > **3** for both mCPBA and in situ-generated PBA (BA with benzaldehyde) systems. In the case of [Fe^II^(PBI)_3_](CF_3_SO_3_)_2_ (**3**), depending on the co-oxidant (H_2_O_2_, mCPBA, and PhIO) used, we have found strong evidence for the formation of μ-1,2-peroxo-diiron(III), acylperoxo-, and iodosylbenzene-iron(III) intermediates, respectively, and their key role in the *Baeyer–Villiger* reaction via A_N_ mechanism. To the best of The authors’ knowledge, this is the second example of a peroxo-mediated catalytic *Baeyer–Villiger* reaction.

## Data Availability

Not applicable.

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
