# Peer review of "Catalytic and Stoichiometric Baeyer–Villiger Oxidation Mediated by Nonheme Peroxo-Diiron(III), Acylperoxo, and Iodosylbenzene Iron(III) Intermediates"

_molecules, 2022, doi:10.3390/molecules27092814_

Round 1

Reviewer 1 Report

In this manuscript, three catalysts, [FeII(PBO)2(CF3SO3)2], [FeII(PBT)2(CF3SO3)2], and [FeII(PBI)3](CF3SO3)2  (PBO = 2-(2’-pyridyl)benzoxazole, PBT = 2-(2’-pyridyl)benzthiazole, PBI = 2-(2’-pyridyl)benzimidazole) have been researched for the Baeyer-Villiger oxidation of cycloketones. Meanwhile, the reaction mechanism also has been studied, but the intermediates of m-1, 2-peroxo-diiron(III) only characterized by UV-Vis spectroscopy. Moreover, there are still some questions in this manuscript. After careful evaluation, I considered this manuscript in current status is not suitable for publication in molecules. Some suggestions are as following:

  1. The description of the structure information of the catalyst in the manuscript is very scarce, and the existing information is difficult to confirm its true structure state. Moreover, identification of intermediate species FeIII(m-1,2-O2)FeIII by UV-Vis spectroscopy alone is not that convincing, other characterization about the structure of FeIII(m-1,2-O2)FeIII also should be add, such as MS,Raman etc.
  2. In this manuscript, relative reactivity of substrates is cyclohexanone > 2-Me-cyclohexanone > 4-Me-cyclohexanone > 4tBu-cyclohexanone > 3-Me-cyclohexanone. The authors attribute it to the increasing bulkiness of the substituents. However, the existing discussion is not enough to explain why the conversion of 2-Me-cyclohexanone and 4-Me-cyclohexanone are larger than 3-Me-cyclohexanone. Meanwhile, the disscusion of 2tBu-cyclohexanone and 3tBu-cyclohexanone also should be added to support the above conclusions.
  3. The speculation on the reaction mechanism in the manuscript is not supported by any relevant characterization technology or design experiments, so it is difficult to determine its rationality. It is suggested to supplement isotope calibration experiments or related characterization techniques to capture the information of intermediate products.
  4. The relative reactivity about catalysts with different N–heterocyclic ligands is [FeII(PBO)2(CF3SO3)2] > [FeII(PBT)2(CF3SO3)2] > [FeII(PBI)3](CF3SO3)2, but there is almost no investigate and discussion about the effect of ligand framework in this paper.
  5. Please check through the manuscript carefully. Some error, such as line 245, Page 10.

Round 2

Reviewer 1 Report

This manuscript has been well improved by the authors and ready for publication.

Reviewer 2 Report

In the revised version of their work, the Authors have cleared the open major issues and several minor points. Therefore, I am glad to recommed this manuscript for publication in its present form.